# Alpha-Lipoic Acid Ameliorates Radiation-Induced Lacrimal Gland Injury through NFAT5-Dependent Signaling

**DOI:** 10.3390/ijms20225691

**Published:** 2019-11-13

**Authors:** Hyuna Kim, Woong-Sun Yoo, Jung Hwa Jung, Bae Kwon Jeong, Seung Hoon Woo, Jin Hyun Kim, Seong Jae Kim

**Affiliations:** 1Department of Ophthalmology, Gyeongsang National University School of medicine and Gyeongsang National University Hospital, Jinju 52727, Korea; landyonel@gmail.com (H.K.); oocee@daum.net (W.-S.Y.); 2Institute of Health Sciences, Gyeongsang National University School of Medicine, Jinju 52727, Korea; jhring@hanmail.net (J.H.J.); blue129j@hanmail.net (B.K.J.); 3Department of Internal Medicine, Gyeongsang National University School of Medicine and Gyeongsang National University Hospital, Jinju 52727, Korea; 4Department of Radiation Oncology, Gyeongsang National University School of medicine and Gyeongsang National University Hospital, Jinju 52727, Korea; 5Department of Otolaryngology-Head and Neck surgery, Dankook University College of Medicine, Cheonan 31116, Korea; lesaby@hanmail.net; 6Biomedical Research Institute, Gyeongsang National University Hospital, Jinju 52727, Korea

**Keywords:** nfat5, dry eye syndrome, alpha-lipoic acid, radiation therapy

## Abstract

Dry eye syndrome related to radiation therapy is relatively common and can severely impair a patient’s daily life. The nuclear factor of activated T cells 5(NFAT5) is well known for its osmoprotective effect under hyperosmolar conditions, and it also has immune-modulating functions. We investigated the role of NFAT5 and the protective effect of α-lipoic acid(ALA) on radiation-induced lacrimal gland (LG) injuries. Rats were assigned to control, ALA only, radiation only, and ALA administered prior to irradiation groups. The head and neck area, including the LG, was evenly irradiated with 2 Gy/minute using a photon 6-MV linear accelerator. NFAT5 expression was enhanced and localized in the LG tissue after irradiation and was related to cellular apoptosis. ALA had a protective effect on radiation-induced LG injury through the inhibition of NFAT5 expression and NFAT5-dependent signaling pathways. Functional radiation–induced damage of the LG and cornea was also restored with ALA treatment. NFAT5 expression and its dependent signaling pathways were deeply related to radiation-induced dry eye, and the condition was improved by ALA treatment. Our results suggest a potential role of NFAT5 and NF-κB in the proinflammatory effect in LGs and cornea, which offers a target for new therapies to treat dry eye syndrome.

## 1. Introduction

Ionizing radiation causes damage in normal tissues located in the field of radiation and induces both functional and morphological changes in glandular tissues, including the salivary gland, thryroid gland, and lacrimal gland (LG) [1,2,3]. Dry eye syndrome is relatively common as a symptom of dose-/volume-dependent acute and late toxicity after radiotherapy [3,4,5,6], the significance of which has been overlooked due to its apparent harmlessness. However, the patients experience complications in daily tasks, such as constant irritation and blurred vision [7], which may in turn lead to psychotic complications [8]. In general practice, the dose to the LG is not constrained (maximum constraint < 40 G y) in radiation therapy [4], and dose reduction is the only considerable option. In the current situation, an alternative method of preventing and treating radiation-induced dry eye is needed, including clarifying the mechanism between radiation and lacrimal dysfunction of the ocular surface.

The nuclear factor of activated T cells (NFAT5), also known as the tonicity response enhancer-binding protein (TonEBP), is a Rel homology transcription factor classically known for its osmosensitive role in regulating cellular homeostasis during states of hypo- and hypertonic stress [9]. Hyperosmolar stress is a key factor of dry eye syndrome, and NFAT5 has been shown to be induced and translocated to the nucleus in ocular tissue cells under hyperosmolar stress. Recent studies suppose this molecule may be an important gene regulation and cell survival factor in the hyperosmolar-stressed ocular surface [10,11], and the tonicity-independent stimulation of NFAT5 is crucial for tissue-specific functions such as enhanced cell survival, migration, and proliferation [9]. To our knowledge, the relationship between NFAT5 and radiation-induced LG injury has not been mentioned.

As a strong antioxidant, α-lipoic acid (ALA) protects tissues from radiation injury in experimental animal models [12,13]. One study found that ALA has a protective effect through lowering lipid peroxidation in children who were exposed to low-dose radiation chronically by the Chernobyl nuclear accident [12]. The effect of ALA on minimizing radiation injury is based on the high reactivity to free radicals that facilitates regeneration of vitamins C and E, elevates tissue levels of glutathione [14], and contributes to the intracellular antioxidant system by scavenging reactive oxygen species, which are involved in the recovery of radiation-induced glandular dysfunction from the early phase [13]. However, the effect of ALA on radiation-induced LG dysfunction has not been examined.

In this study, we investigated the role of NFAT5 in radiation-induced LG injury and the protective effect of ALA on the LG and the mechanisms.

## 2. Results

### 2.1. NFAT5 Expression during the LG Injury Progression to Radiation

Previous reports have revealed the associations between the pathogenesis of ocular damage in dry eye syndrome and cataracts and NFAT5 expression [10,11,15,16]. However, the time profile of alterations in NFAT5 expression during the course of LG injury progression after radiation has not been examined. To elucidate the relationship between NFAT5 expression and radiation-induced LG injury, we first evaluated tissue levels of NFAT5 expression at 2, 6, 8, and 12 weeks after radiation. We observed a significant increase in the expression of NFAT5 in the LG at 2, 6, and 8 weeks after radiation, but the expression was decreased in a time-dependent manner (Figure 1). We also found normalization of NFAT5 expression at 12 weeks after radiation. This result suggests that NFAT5 might be involved in radiation-induced LG injury. Due to the high expression of NFAT5 in the LG two weeks after radiation, we focused our attention on this time point thereafter.

### 2.2. Effect of ALA on Radiation-Induced NFAT5 Expression in the LG

To confirm if NFAT5 is involved in radiation-induced LG injury, structural changes and localization of NFAT5 expression were examined in the LG after radiation. As shown in Figure 2A, unaltered acini and intercalary ducts were observed in the control and ALA-only groups. However, multiple tiny and large vacuoles in the cytoplasm of the acinar cells and the nuclei periphery were seen in the RT group. Of note, NFAT5 expression was markedly localized in the nuclei of injured acinar cells in the RT group, as was radiation-induced structural damage. These positive signals for NFAT5 were well correlated with NFAT5 expression from tissue lysates (Figure 1). We are convinced that NFAT5 expression must be involved in radiation-induced LG injury. We have already reported the protective effects of ALA on various tissue injuries after radiation [13,17,18,19]. We asked whether ALA could protect radiation-induced LG injury. Figure 2 indicates that ALA ameliorates histological changes (ALA + RT in Figure 2A–C) and NFAT5 expression (ALA + RT in Figure 2D,E) in the LG after radiation.

### 2.3. Effect of ALA on Radiation-Induced Apoptosis in the LG

To test whether ALA can also protect against radiation-induced cell death in the LG as well as structural damage, cleaved caspase-3 expression and TUNEL staining was performed. Cleaved caspase-3 expression, one of the markers for apoptotic cell death, was significantly increased in the RT group, and the expression declined after ALA treatment (ALA + RT; Figure 3A,B). TUNEL-positive signals were observed in acinar cells from the RT group, and the signals were also decreased in the ALA-treated RT group (ALA + RT; Figure 3C). We next examined whether NFAT5 plays a crucial role in apoptosis of the LG after radiation. First, we performed double staining for NFAT5 and TUNEL and found, two weeks after radiation, markedly increased double-positive signals in the LG in the RT group. Furthermore, the signals were significantly decreased in rats subjected to radiation and injected with ALA. These results indicate that NFAT5 expression in the LG plays an important role in cell death and that ALA ameliorates NFAT5-involved cell death in the LG after radiation.

### 2.4. Suppression of NFAT5-Dependent Signaling Pathway by ALA Treatment in the LG

Due to the fact that NF-κB expression is primarily induced by NFAT5 [11,20,21,22,23], we speculated that ALA can also suppress the radiation-induced increase in NF-κB expression in the LG. Increased levels of phosphorylated NF-κB expression by radiation was significantly reduced in the ALA-treated group (ALA + RT; Figure 4). Mitogen-activated protein kinases (MAPKs) are also known to be a specific factor in the NFAT5-dependent signaling pathway [10,21,24,25]. ALA attenuated the increase in phosphorylated p38 and pJNK expression and the decrease in phosphorylated ERK by radiation (Figure 4). These data suggest that ALA might be protecting the LG against radiation through alleviation of the NFAT5-dependent signaling pathway.

### 2.5. Protective Effects of ALA in Radiation-Induced LG Injury

To test whether NFAT5 functionally contributes to, and is necessary for, gland injury following radiation, we examined the expression of AQP-5 [26]. The AQP-5 expression level was significantly decreased by radiation, whereas administration of ALA restored the expression to basal levels (Figure 5). This result, together with that obtained from Figure 2 and Figure 3, suggests that ALA could contribute to restoring radiation-induced LG dysfunction following radiation.

### 2.6. Effect of ALA on Corneal Fluorescein Staining and Integrity of the Corneal Epithelium

Finally, we asked whether ALA can also protect radiation-induced injury in the cornea, which is directly related to dry eye symptoms, not only affecting the LG. To verify this, fluorescein staining was employed to the corneas. The fluorescein staining of the cornea was significantly increased by radiation (Figure 6A,C). However, the staining score of the ALA-treated RT group was markedly decreased and differed statistically from the control and ALA-only groups. To investigate the damage to corneal structure, the corneal sections were stained with H&E. Figure 6B,D demonstrates the regular structural pattern of the corneal epithelium, Bowman’s layer, and stroma in the control and ALA groups in contrast to the RT group, which showed clear thinning of the epithelium; the corneal structure was shown to be improved with ALA treatment.

## 3. Discussion

NFAT5 expression was increased in the nuclei of injured acinar cells of the LG after irradiation and was related to cellular apoptosis. ALA had a protective effect on radiation-induced LG injury by inactivation of NFAT5-dependent signaling pathways, as well as inhibition of NFAT5 expression. The effect of ALA not only was limited to protect histologic and biochemical changes, as well as the dysfunction of LG, but the corneal epithelium was also restored with ALA treatment. The present study suggests that NFAT5 might be involved in radiation-induced LG and that ALA can ameliorate the injury through alleviation of the NFAT5-dependent signaling pathway.

In the recent prospective study, 46% of patients experienced dry eye symptoms one month after radiotherapy. These symptoms can begin immediately after therapy and be sustained for up to six months [3]. Dry eye syndrome after radiotherapy results not only in acute discomfort, but also in chronic disease during the patient’s remaining years. In intensity-modulated radiotherapy for sinonasal tumors, the lacrimal acinar changes were found to exceed those in the orbital structures [27]. The pathogenesis of radiation-induced dry eye syndrome is unclear, and the main cause of dry eye is still difficult to solve (i.e., lack of tear production). Moreover, most treatments are limited to temporary relief symptoms. Therefore, it may be necessary to stimulate or restore tear production potential using clinically relevant agents that have been verified for clinical use. From this point of view, the current results indicate that ALA may help patients with radiation-induced dry eye symptoms by preventing and restoring LG dysfunction after irradiation.

Hyperosmolar stress is one of the key mechanisms of all types of dry eye syndrome [28]. NFAT5 was initially identified from the kidney medulla and it is activated by osmotic shock [21]. Through extracellular hypertonicity, NFAT5 induces the expression of osmoprotective genes, such as ion transporters, aldose reductase, and heat shock protein 70, to neutralize the osmotic gradient [21]. NFAT5 is known to increase cell viability through this process in various tissues [15,16,29,30]. On the other hand, NFAT5 induces cellular apoptosis and leads to tissue damage in tonicity-independent circumstances [21]. The present study belongs to the latter process, as NFAT5 expression was closely related to radiation-induced cellular apoptosis in the LG. Therefore, we speculate that the dysfunction of the LG could be a major cause of radiation-induced dry eye syndrome.

We reported that UVB radiation stimulated the interaction between NFAT5 and NF-κB in human lens epithelial cells [20]. UVB light triggers activation and nuclear translocation of NFAT5, and it promotes the interaction with NF-κB p65. NF-κB plays multiple roles in cell survival, development, apoptosis, immunity, and inflammatory responses [31,32]. Roth et al. [23] showed an interaction between NFAT5 and NF-κB p65, and that NF-κB activity is enhanced in association with NF-κB/NFAT5 complexes. Zhou et al. [22] revealed that elevated reactive oxygen species levels contribute to the activation of NFAT5. In agreement with these findings, the present study shows that radiation stress changed the level of phosphorylated NF-κB and MAPKs (p38, JNK, ERK) in LG tissue. The MAPK pathways play an important role by mediating cellular functions, including apoptosis and inflammation [24,25]. Activation of p38 and JNK also contributes to inducing inflammatory cytokine and chemokines, which amplify the inflammatory process at the injury site [33,34]. The NF-κB, p38, and JNK levels were positively related to NFAT5 expression after radiation injury. Of note, ALA treatment minimized these changes. Andrade et al. [35] reported that ALA restored tear production in animals with dry eye and that the effect of ALA comes from the alteration of the metabolism of reactive nitrogen species, causing increased activity of lacrimal peroxidase. In our results, ALA significantly reduced TUNEL-positive signals in LG ductal cells and acinar cells of irradiated rats (Figure 3). All told, elevated NFAT5 expression by radiation induced NF-κB, p38, and JNK expression and the activation in the LG. This elevated expression may lead to cellular apoptosis, which was preventable by ALA treatment. No studies have reported the relationship between ALA and NFAT5-induced signal pathways. We believe that ALA reduces the NFAT5–NK-κB axis related to inflammation in the LG after radiation and that these effects of ALA might ameliorate radiation-induced dry eye syndrome.

The pathogenesis of radiation-induced dry eye syndrome is still unclear and may cause LG damage and destruction of ocular structures including cornea, conjunctiva, and minor lacrimal glands. We showed the LG damage could be a main cause of dry eye syndrome after radiation in this study, although the contribution of damage in ocular apparatus was not fully checked biochemically and histologically. However, NEI scores and corneal epithelial heights were improved after ALA treatment, without local treatment for cornea, conjunctiva, or minor LGs. Also, the actual tear volume change with ALA pretreatment was not checked directly by experiments. We tried to measure the functional improvement of the corneal surface by examining corneal fluorescein staining scores and corneal epithelial thickness changes, but still, the exact volume change measurement is requirement to verify glandular functional restoration. Finally, the radiation-induced dry eye rat model is not a standardized dry eye model. Fine adjusting of radiation and the pretreatment method is crucial for adapting this result to humans. In addition, this methodological problem could be minimized and advanced by using our experience in dealing with various tissues (salivary glands, oral mucosa, thyroid, etc.) [13,17,19]. Therefore, the effectiveness of this model as a preclinical dry eye syndrome model should be cleared up, and we are trying to define the most methodologically stable and effective model in a further study.

## 4. Materials and Methods

### 4.1. Ethics Statement

This study was approved on Oct. 31, 2016 by the Gyeongsang National University Institutional Animal Care and Gyeongsang National University Institutional Ethics Committee (GNU-16031-R0059).

### 4.2. Radiation Exposure

We assigned male *Sprague–Dawley* rats (230–250 g; Koatech Inc., Peongtaek, Korea) to the following groups control, radiation alone (RT), ALA administration before irradiation (ALA + RT), and ALA administration alone (ALA). We administered ALA (100 mg/kg intraperitoneally; Bukwang Pharmaceutical Co., Seoul, Korea) either 24 h or 30 min before irradiation, and we chose the dose and frequency based on previous studies [13,17,19]. The head and neck area was evenly irradiated with 2 Gy/min (total dose, 18 Gy) using a photon 6 MV linear accelerator (21EX; Varian, Palo Alto, CA, USA). A 1.5 cm Bolus was positioned above the head and neck to provide adequate buildup and facilitate even radiation distribution. Each rat was exposed to a single dose of radiation and was sacrificed 2, 6, 8, or 12 weeks after radiation. All experiments were performed in triplicate with *n* = 7 animals in each group.

### 4.3. Immunoblotting

The samples were obtained from extraorbital lacrimal glands, which were removed bilaterally for histopathological evaluations and immunoblotting 2, 6, 8, and 12 weeks after irradiation. The tissues were homogenized in RIPA buffer (#89900. Thermo scientific. Waltham, MA, USA). Amounts of protein were measured by BCA assay kit (Pierce, Rockford, IL, USA) according to the manufacturer’s protocol. Proteins (50 µg) were loaded and electroblotted. The blots were probed with primary antibodies against monoclonal anti-NFAT5 (Santa Cruz Biotechnology, Santa Cruz, CA, USA), polyclonal anti-cleaved caspase-3, anti-phosphorylated NF-κB, p38, pJNK, and pERK purchased from Cell Signaling Technology (Danvers, MA, USA) at 4 °C overnight. The primary antibody was visualized by a secondary antibody and an ECL kit (Amersham Pharmacia Biotech, Piscataway, NJ, USA).

### 4.4. Double Immunofluorescence Staining

To characterize NFAT5 and apoptosis-positive cells, double immunofluorescence was performed on the lacrimal glands. Deparaffinization and antigen retrieval were performed. Nonspecific antibody binding was blocked in PBS with 0.1% normal donkey serum (Vector Laboratories, Burlingame, CA, USA) and 0.3% Triton X-100 (Sigma, St. Louis, MO, USA) for 45 min. Sections were briefly placed in phosphate-buffered saline (PBS), boiled in sodium citrate (10 mM, pH 6.0) for 10 min, and cooled at 4 °C for 20 min. The sections were incubated with monoclonal anti-NFAT5 overnight 4 °C. After washing with PBS, the sections were incubated with secondary Alexa594-conjugated rabbit antimouse IgG for 1 h at room temperature. Then, apoptosis in LG tissues was determined by terminal deoxynucleotidyl transferase dUTP nick end labeling (TUNEL) assay using an ApopTag Plus In Situ Apoptosis Kit (Chemicon Int., Temecula, CA, USA). Double-positive cells for NFAT5 and TUNEL were photographed under 200× magnification.

### 4.5. Immunohistochemistry

After deparaffinization, the sections were incubated with primary antibodies against monoclonal anti-NFAT5 and polyclonal anti-AQP5 (Abcam, Cambridge, UK), followed by biotin-conjugated secondary IgG (diluted 1:200; Vector Laboratories, Burlingame, CA, USA), avidin–biotin–peroxidase complex (ABC Elite Kit; Vector Laboratories), and DAB. Next, we visualized the sections by light microscopy and captured and analyzed digital images.

### 4.6. Corneal Fluorescein Staining

Corneal fluorescein staining was done for examined end organ change of dry eye syndrome, using the National Eye Institute (NEI) grading system, giving a score from 0 to 3 (0  =  normal and 3 = severe) for each of the five areas (superior, nasal, central, inferior, temporal) of the cornea [18]. After loading ten microliters of 1% fluorescein into the lateral conjunctival sac of the rats, 500 μL of normal saline was applied to corneal surface to it wash out. The epithelial defects of the corneal surface were stained with fluorescein, this staining was examined by slit lamp biomicroscope (SL 120, Carl Zeiss Meditec AG, Jena, Germany).

### 4.7. Histology for Corneal Epithelium

The eyeballs of rats were surgically enucleated and fixed in 10% formalin. The tissues were embedded in paraffin and cut to 5 μm. For the evaluation of corneal tissue injury, the sections were stained with hematoxylin and eosin (H&E) and photographed with a virtual microscope (Nikon, Minato, Japan). Corneal epithelium height was measured in five sections of each eye using NIS Elements BR3.2 (Nikon, Minato, Japan).

### 4.8. Statistical Analysis

Statistical analyses were performed using GraphPad Prism software (version 8.0; GraphPad Software Inc., La Jolla, CA, USA). The Mann–Whitney U test was used to examine differences between all groups. In all analyses, *p* < 0.05 was taken to indicate statistical significance.

## 5. Conclusions

Radiation exposure to the head and neck induced NFAT5 expression and its dependent signaling pathways in the LG, which is deeply related to radiation-induced dry eye. However, the gland injury was improved by ALA treatment. This study suggests that NFAT5 might be one of the targets in the treatment of radiation-induced dry eye syndrome. Furthermore, ALA could be a preventive/treatment option for other patients suffering from dry eye symptoms.

## Figures and Tables

**Figure 1 ijms-20-05691-f001:**
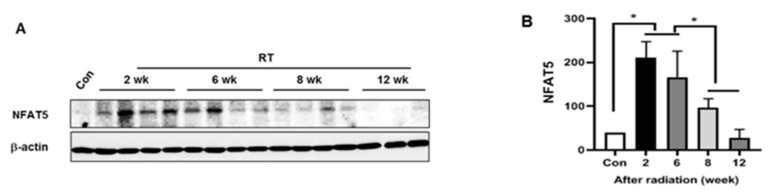
The nuclear factor of activated T cells (NFAT5) is differentially expressed by times in the lacrimal gland after radiation. (**A**) Lacrimal gland expression of NFAT5 in control and injured glands 2, 6, 8, or 12 weeks after radiation. β-actin was used as internal control. (**B**) Signal density of NFAT5 expression level in all groups. Data are expressed as mean ± SEM. * *p* < 0.05 versus each marked group. Con, control. RT, radiation.

**Figure 2 ijms-20-05691-f002:**
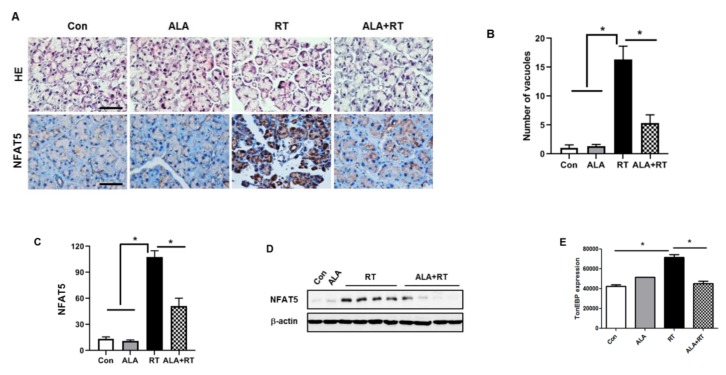
α-lipoic acid (ALA) decreased radiation-induced structural changes and NFAT5 expression in the lacrimal gland. (**A**) Histopathological changes and immunohistochemical staining micrographs show NFAT5 expression. (**B**) Pathological scoring is examined by number of acinar cells with vacuoles. (**C**) Positive signal density of NFAT5 expression level in all groups. (**D** and **E**) Lacrimal gland expression of NFAT5 in all groups 2 weeks after radiation. Signal density of NFAT5 expression level in all groups. * *p* < 0.05 versus each marked group. Con, control. ALA, alpha-lipoic acid. RT, radiation. ALA + RT, ALA and radiation. Scale bar, 50 μm.

**Figure 3 ijms-20-05691-f003:**
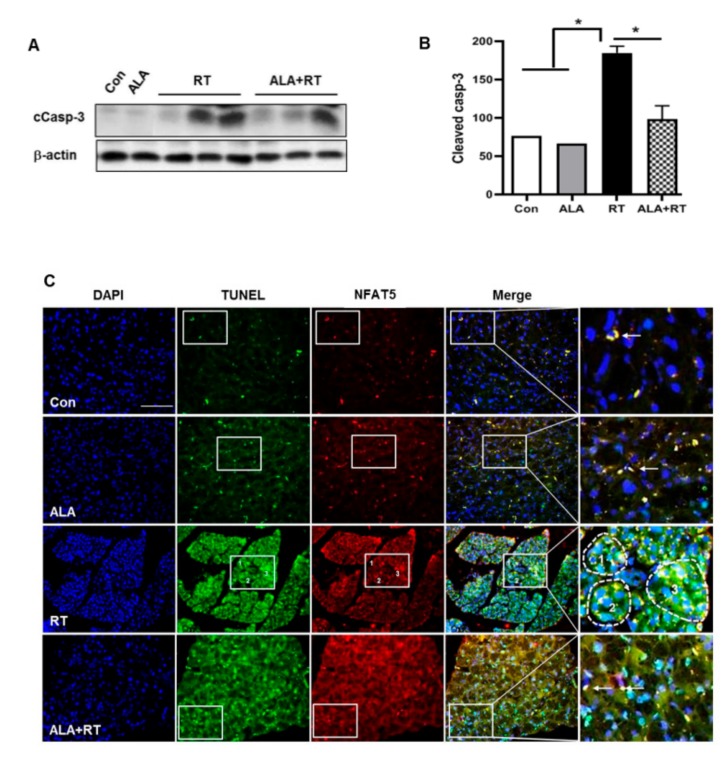
ALA ameliorates radiation-induced apoptotic cell death in the lacrimal gland. (**A** and **B**) Lacrimal gland expression of cleaved caspase-3 in all groups, 2 weeks after radiation. Signal density of cleaved caspase-3 expression level in all groups. (**C**) NFAT5 expression and apoptosis. Boxed areas are enlarged and presented in the right column. Arrows indicate positive signals. Dot lines in RT group indicate abundant positive signals. * *p* < 0.05 versus each marked group. Con, control. ALA, alpha-lipoic acid. RT, radiation. ALA + RT, ALA and radiation. Scale bar, 100 μm.

**Figure 4 ijms-20-05691-f004:**
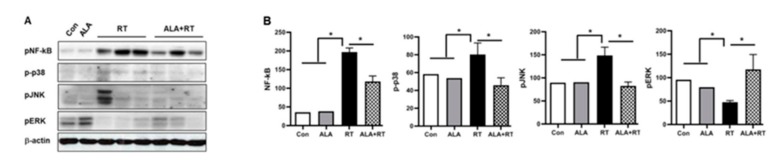
ALA inactivates the NFAT5-involved signaling pathway by radiation in the lacrimal gland. (**A**) Lacrimal gland expression of NF-κB and MAPKs (p-p38, pJNK, and pERK) in all groups 2 weeks after radiation. (**B**) Signal density of each expression level in all groups. * *p* < 0.05 versus each marked group. Con, control. ALA, alpha-lipoic acid. RT, radiation. ALA + RT, ALA and radiation.

**Figure 5 ijms-20-05691-f005:**
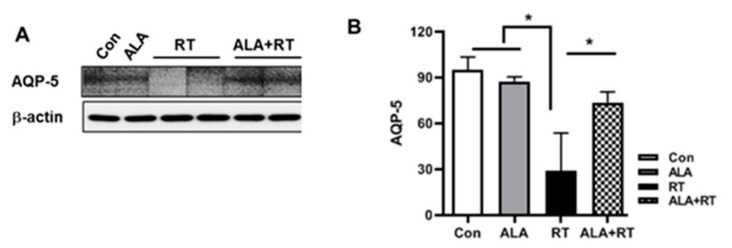
ALA restores radiation-induced lacrimal gland dysfunction. (**A**) The lacrimal gland tissue lysates were examined by immunoblot with antibodies against AQP-5. (**B**) Quantified analysis was performed by densitometry. Data are mean ± SEM. * *p* < 0.05 versus each marked group. Con, control. ALA, alpha-lipoic acid. RT, radiation. ALA + RT, ALA and radiation.

**Figure 6 ijms-20-05691-f006:**
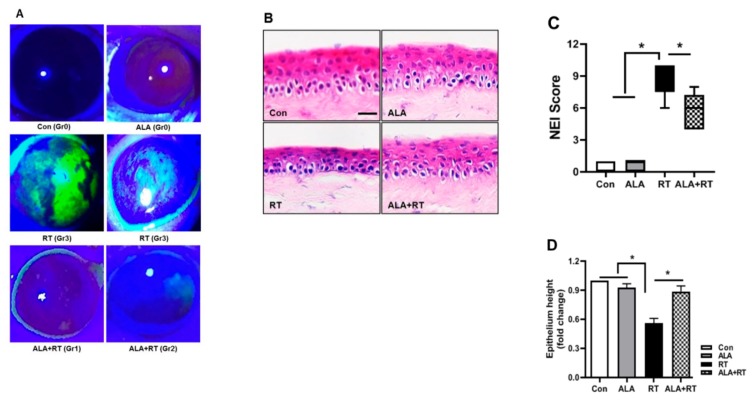
ALA ameliorates radiation-induced corneal fluorescein staining. (**A**) Fluorescent slit-lamp photographs of the eyes of all groups. (**B**) Corneal epitheliums of each groups stained with H&E. Scale bar, 25 μm (**C**) Corneal fluorescein grading score of each group. (**D**) Corneal tissue injury was scored by epithelium height. The quantitative data are presented as means ± SEM. * *p* < 0.05 versus each marked group. Con, control. ALA, alpha-lipoic acid. RT, radiation. ALA + RT, ALA and radiation.

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
