# Peer review of "Alpha-Lipoic Acid Ameliorates Radiation-Induced Lacrimal Gland Injury through NFAT5-Dependent Signaling"

_ijms, 2019, doi:10.3390/ijms20225691_

Round 1

Reviewer 1 Report

General Comments

In the manuscript IJMS-636942 “Alpha-lipoic acid ameliorates radiation-induced lacrimal gland injury through NFAT5-dependent signaling”, the authors examined the role of NFAT5 and a-lipoic acid on radiation-induced lacrimal gland injury. 

Overall I thought this was a nice presentation of the data and I have no issue with it.  My only issue is with the Materials and Methods

Minor Comments

Section 4.3 needs significantly more information. Please specific what tissues, how were the tissues obtained?  What is the lysis buffer?   How was total protein determined?  What mM concentration was the sodium citrate in 4.4? I realize the methods section needs to be kept short and succinct, however, future researchers need to be able to reproduce the experiments. Please do a careful review of this section and make sure the experiments are described adequately enough. 

Author Response

Thank you for your careful comment. We apologized for brief description on the section of Methods.

4.3 Immunoblotting

The samples were obtained from extraorbital lacrimal gands which were removed bilaterally for histopathological evaluations and immunoblotting on 2, 6, 8, and 12 weeks after irradiation. The tissues were homogenized in RIPA buffer (#89900. Thermo scientific. Waltham, MA, USA). Amounts of protein were measured by BCA assay kit (Pierce, Rockford IL, USA) according to the manufacturer’s protocol. .

4.4 Double immunofluorescence staining

To characterize NFAT5 and apoptosis-positive cells, double immunofluorescence was performed on the lacrimal glands. Deparaffinization and antigen retrieval were performed. Non-specific antibody binding was blocked in PBS with 0.1% normal donkey serum (Vector Laboratories) and 0.3% Triton X-100 (Sigma) for 45 min. Sections were briefly placed in phosphate-buffered saline (PBS), boiled in sodium citrate (10 mM, pH 6.0) for 10 min, and cooled at 4°C for 20 min. The sections were incubated with monoclonal anti-NFAT5 overnight 4°C. After washing with PBS, the sections were incubated with secondary Alexa594-conjugated rabbit anti-mouse IgG for 1 h at room temperature. Then, apoptosis in LGs tissues was determined by terminal deoxynucleotidyl transferase dUTP nick end labeling (TUNEL) assay using an ApopTag Plus In Situ Apoptosis Kit (Chemicon Int., Temecula, CA, USA). Double-positive cells for NFAT5 and TUNEL were photographed under 200× magnification

Reviewer 2 Report

page2 at line 49, NFAR5 should be revised to TonEBP. page2 at Figure 1 Legend, "b" should be revised to "β". If you discuss about the tear volume in Discussion, you need to examine tear volume as well as CFS scoring.

Author Response

Thank you for your careful comments.

Point 1 : page2 at line 49, NFAR5 should be revised to TonEBP.

Response 1: Changed as mentioned.

Point 2 : page2 at Figure 1 Legend, "b" should be revised to "β".

Response 2: Changed as mentioned.

Point 3 : If you discuss about the tear volume in Discussion, you need to examine tear volume as well as CFS scoring. 

Response 3: As the reviewer pointed out, the exact volume change measurement is requirement to verify glandular functional restoration. We tried to the functional improvement of corneal surface by examine corneal fluorescein staining score and corneal epithelial thickness change, without tear volume increase, these definite improvements are hard to appeared. But it is still a limitation that pressing for serious consideration; we added these comments in the discussion section (Page ; line - ). The tear volume issue was described in the discussion but it was limited within the scope of needs for this study, and the effects of ALA on the references. And the increased tear volume is a one of essential part of the factor initiating dry eye syndrome, we carefully consider the discussed words remained. We are planning for consecutive experiment, including measuring the exact tear volume. 

We deeply appreciate your time and effort for our manuscript.

Reviewer 3 Report

Radiation exposure to the head and neck induced NFAT5 expression and its dependent signaling pathways in the lacrimal gland, which is deeply related to radiation-induced dry eye. In this study, the gland injury was improved by α-lipoic acid(ALA) treatment. This study suggests that NFAT5 might be one of the targets in the treatment of radiation-induced dry eye syndrome. In the future, ALA may be a preventive/treatment option for other patients suffering from dry eye symptom.

Author Response

Thank you for careful comment. As you commented, we are expect the ALA and NFAT5 related pathway could be a new treatment target of dry eye syndrome, whatever the cause is. We are planning for the consecutive study with the more advanced causable effects of NFAT5 induced pathway in the other ocular disease. We deeply appreciate your time and effort for our manuscript.